# Silicon Photonic Micro-Transceivers for Beyond 5G Environments

**Kazuhiko Kurata [1,\*], Luca Giorgi [2], Fabio Cavaliere [2], Liam O'Faolain [3,4], Sebastian A. Schulz [3], Kohei Nishiyama [5], Yasuhiko Hagihara [1], Kenichiro Yashiki [1], Takashi Muto [1], Shigeru Kobayashi [1], Makoto Kuwata [1] and Richard Pitwon [1,3,6,\*]**

[1] AIO Core Ltd., 47-12-301, Sekiguchi 1-Chome, Bunkyo-ku, Tokyo 104-0014, Japan; y-hagihara@aiocore.com (Y.H.); k-yashiki@aiocore.com (K.Y.); t-muto@aiocore.com (T.M.); s-kobayashi@aiocore.com (S.K.); m-kuwata@aiocore.com (M.K.)

[2] Ericsson Research, Via Moruzzi 1 c/o CNR, Ericsson, 56124 Pisa, Italy; luca.giorgi@ericsson.com (L.G.); fabio.cavaliere@ericsson.com (F.C.)

[3] School of Physics and Astronomy, University of St Andrews, North Haugh, St. Andrews KY16 9SS, UK; William.Whelan-Curtin@mtu.ie (L.O.); sas35@st-andrews.ac.uk (S.A.S.)

[4] Centre for Advanced Photonics and Process Analysis (CAPPA), Munster Technology University, T12P928 Cork, Ireland

[5] I-PEX Inc., Machida ST BLD., 1-33 Morino, Tokyo 104-0014, Japan; nishiyama.kohei@i-pex.com

[6] Resolute Photonics, Northover House, 132a Bournemouth Road, Chandlers Ford, Eastleigh SO53 3AL, UK

\* Correspondence: k-kurata@aiocore.com (K.K.); rpitwon@resolutephotonics.com (R.P.)

**Abstract:** Here, we report on the design and performance of a silicon photonic micro-transceiver required to operate in 5G and 6G environments at high ambient temperatures above 105 °C. The four-channel "IOCore" micro-transceiver incorporates a 1310 nm quantum dot laser system and operates at a data rate of 25 Gbps and higher. The 5 × 5 mm micro-transceiver chip benefits from a multimode coupling interface for low-cost assembly and robust connectivity at high temperatures as well as an optical redundancy scheme, which increases reliability by over an order of magnitude.

**Keywords:** silicon photonics; 5G; 6G; co-packaged optics; data centers; integrated photonics; micro-transceiver; high-performance computer (HPC)

## 1. Introduction

The conditions required to accommodate the increasing bandwidth densities in 5G+, hyperscale data centers and high-performance computer (HPC) environments are becoming more extreme. Given the requirement that larger numbers of optical transceiver I/O ports are moved closer to the signal source chip in the system (e.g., ASIC or FPGA) in embedded or co-packaged optical assemblies, it follows that transceivers will need to be packed more tightly into very confined spaces. Consequently, transceivers will need to operate at higher ambient temperatures than those to which commercial transceivers have historically been subjected, in many cases exceeding 105 °C as we will outline later on. In order to be deployed in these hyperscale environments, optical transceivers will need to provide higher aggregate bandwidths, a reduced footprint and higher reliability at operating temperatures exceeding 100 °C. The design and performance of a silicon photonic micro-transceiver are presented, which can operate under high ambient temperature regimes exceeding 105 °C. The quad (4 Tx + 4 Rx) silicon photonic micro-transceiver "IOCore" incorporates a 1310 nm Fabry-Perot Quantum Dot laser, the output of which is modulated at data rates of 25, 32 and 50 Gbps per lane. The compact 5 × 5 mm micro-transceiver chip features a multimode optical coupling interface and an optical redundancy circuit. Together, these enable robust, low-cost operation and connectivity at high temperatures and increase reliability by over an order of magnitude compared to non-redundant integrated laser schemes.

The first generation of 100G silicon photonics micro-transceiver comprised four bidirectional channels operating at 25 Gbps [1]. AIO Core collaborated with partners of the European H2020 COSMICC project [2] to demonstrate its first generation of silicon photonics micro-transceivers on mezzanine test cards [3], which have recently been standardized by the IEC [4]. The COSMICC hyperscale optical interconnect ecosystem demonstration platform "Aurora" was exhibited at ECOC 2019 in Dublin in September 2019 (Figure 1).

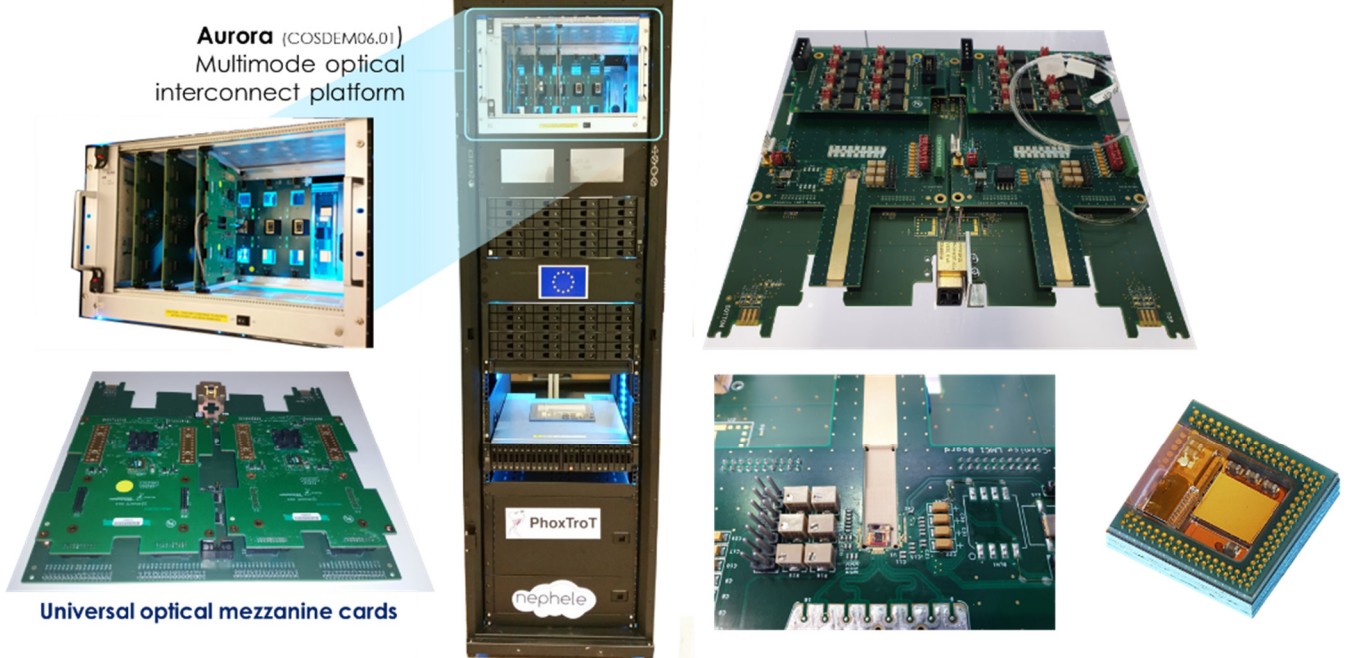

**Figure 1.** Hyperscale rack-scale demonstrator with H2020 COSMICC "Aurora" platform and test daughtercards with I/O core mezzanine cards connected [5–7].

Silicon photonics-based transceivers were initially developed for data center interconnects (DCIs), where the high requisite volumes of optical transceivers enabled critical investment in new technologies.

Silicon photonics has evolved over the past five years from a high-cost margin telecom technology to a commoditized datacom transceiver technology capable of transmitting bit rates of hundreds of Gbps at transmission costs of an ever decreasing fraction of a US dollar per Gbps and a power consumption of less than 15 picojoules per bit. A relevant example is the specification recently published by the Co-Packaged Optics Collaboration [8] of a 3.2 Tbit/s co-packaged transceiver module [9]. The advent of 5G has given rise to the requirement for high data rates in the fronthaul segment [10], and this trend is expected to increase with the introduction of advanced antenna systems (AAS) [11] and radio transmission in the millimeter wave range [12]. However, the interconnection speed remains lower in radio communications than in DCI: 100 Gbps links are being introduced in the current generation, with the requirement expected to increase to 400 Gbps within the next few years. A possible scenario is depicted in Figure 2.

Two ASICs (depicted in blue in Figure 2)—for example, the ASIC used for digital baseband processing functions—are connected to each of four 400 Gbps optical modules through four 50 Gbps electrical lanes. Each optical module receives inputs from both ASICs in a dual star redundant configuration for protection and computation offload purposes. These four optical modules connect to another four modules across an optical link, the length of which can range from a few centimeters for intra-board optical interconnects to hundreds of meters for split radio equipment. Each of the optical modules on the right-hand side is connected to an ASIC (green), which could, in an advanced antenna system, be a radio frequency processing ASIC connected to an RF antenna element. This architecture

can be easily scaled to higher capacities by adding further optical modules or to connect a higher number of antenna elements, thus accommodating various product variants with different bandwidths and antenna array sizes. Radio equipment generally requires a lower aggregate capacity than DCIs [13] but also the capability to scale up capacity and power consumption more effectively. Moreover, even if the transceiver volumes in radio are comparable or even higher than in DCI applications, the target cost is lower. This makes it more difficult for transceiver manufacturers to assume the risk of investing in new technologies while maintaining a sustainable business model. The adoption of standardized solutions and the definition of a limited number of implementation options can definitely mitigate this issue avoiding excessive market fragmentation.

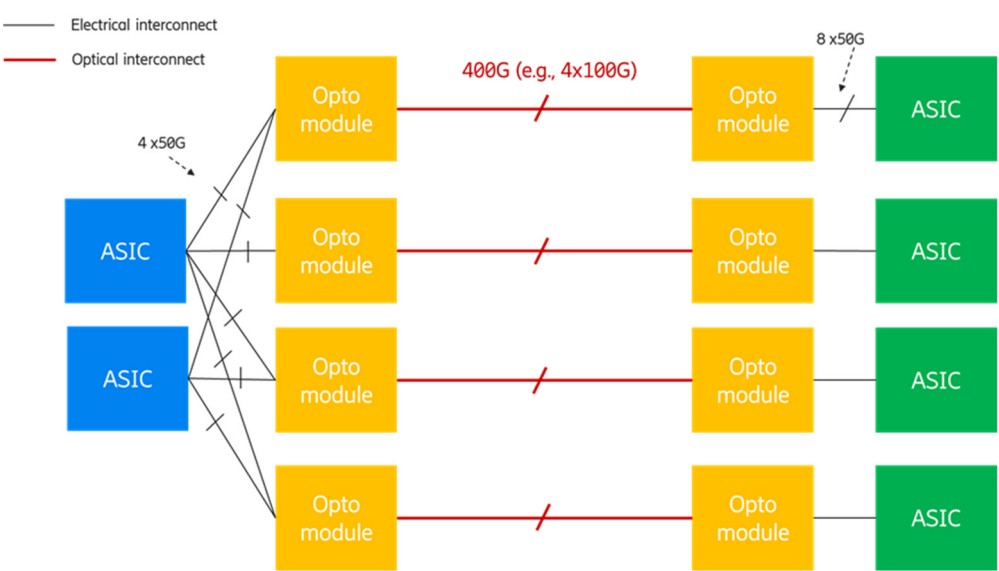

**Figure 2.** Possible interconnection scenario in radio systems.

Power consumption constraints are also more demanding in radio access networks (RANs) than in DCIs. Data center equipment operates indoors in environments with controlled temperature and humidity. In addition, active cooling systems, e.g., line fans, are commonly implemented, with more advanced solutions such as liquid cooling gaining prominence. Constraints in weight, power supply and installation practices make these solutions impractical or not feasible for radio equipment deployed outdoors or at the top of an antenna pole. Therefore, current RANs require transceivers capable of operating in the so-called industrial temperature (I-Temp) range, from $-40$ to $85\,^\circ$C, but the increase in component density in radio boards, due to miniaturization and increase in processing capabilities, is pushing the upper bound of the temperature range above $100\,^\circ$C. While silicon photonic devices, such as modulators and photodetectors, are tolerant to high temperatures, lasers are very sensitive to them. Transceiver operation beyond $85\,^\circ$C requires the use of either external laser sources, placed outside the transceiver [14], or quantum dot lasers, which are very promising to withstand high temperatures [15]. Variations in temperature are particularly problematic for wavelength division multiplexing (WDM) systems.

In 5G (and beyond) radio systems operating in the millimeter range, radiating antenna elements and photonic interconnects will be co-integrated in a single hardware unit. This will not only increase the operational temperature but also calls for miniaturized devices—for example, co-packaged transceivers integrated into single multi-chip modules with digital integrated circuits (ICs) and analogue radio frequency (RF) front-ends. This scenario needs more accurate fiber attachment techniques for silicon photonic chips [16] as well as a radical decrease in the power consumption, from the current tens of picojoules per bit to a few picojoules per bit [17]. The IPSR-I integrated photonics roadmap for transceivers [18] makes clear that photonic packaging technology must improve substantially to achieve better energy efficiency. Integrated photonic solutions for radio applications can bene-

fit from the current DCI technology, such as single-mode silicon photonic transceivers; however, operating temperatures, power consumption and cost margins will be much more challenging in 5G+ environments going forward and will require new solutions. In the next section, we provide a detailed technical description of a multimode silicon photonics micro-transceiver designed to operate in high-temperature environments with an integrated laser for lower overall power consumption that can meet the stringent cost targets of future 5G+ environments due to multimode packaging.

## 2. Silicon Photonics Micro-Transceiver with Multimode Interface

The "IOCore" silicon photonics micro-transceiver chip operates in the 1310-nanometer wavelength regime (O-band) with on-off keying (OOK) modulation data rates per lane of 25, 32 and 50 Gbps. An IOCore chip incorporates four optical transmission and four optical receiver channels under a proprietary multimode optical waveguide or "optical pin" array interface [1] on a 5 × 5 mm footprint (Figure 3).

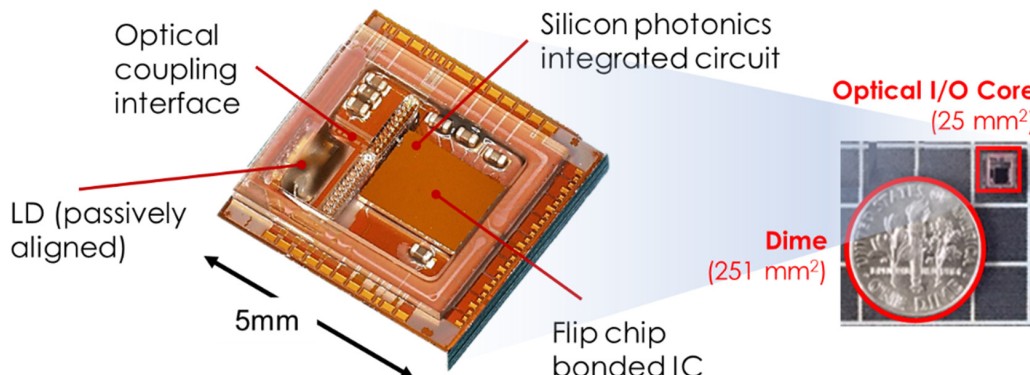

**Figure 3.** IOCore micro-transceiver footprint of 5 × 5 mm [3].

The IOCore transceiver serves as a transceiver building block that can be assembled into a variety of different standard or bespoke packages to address diverse high-volume, high-temperature and space-restricted applications including co-packaged optics in 5G/6G, hyperscale data centers, HPCs and automotive systems.

We will now introduce the overall structure of the IOCore transceiver, before going on to describe its individual components in more detail.

### 2.1. Functional Overview of Transceiver Operation

Figure 4 shows the schematic cross-section and optical interfaces of the IOCore micro-transceiver.

Figure 4b shows a top view of the complete IOCore transceiver chip, which is split into a transmitter section on the left-hand side and a receiver section on the right-hand side.

In the transmitter section (Figure 4d), an edge emitting a Fabry-Perot Quantum Dot laser diode is passively aligned within a compliant recess in the silicon substrate such that it couples 1310-nanometer continuous wave (CW) light into four single-mode silicon waveguides in the substrate. Each waveguide conveys the CW light to a Mach–Zehnder interferometer (MZI). High-speed differential electronic signals passed to the chip through electrical signal pads drive the MZI modulator circuit generating the corresponding modulated 1310-nanometer optical signal, which is then coupled out of the chip through a vertical grating coupler.

The vertical grating coupler couples the optical signal out of the silicon photonic substrate at an angle of 82° from the plane of the silicon substrate into a short untapered optical waveguide called an "optical pin". Figure 4a shows a photo of a proprietary array of twelve optical pins on the transmitter section, of which only four are used in this design.

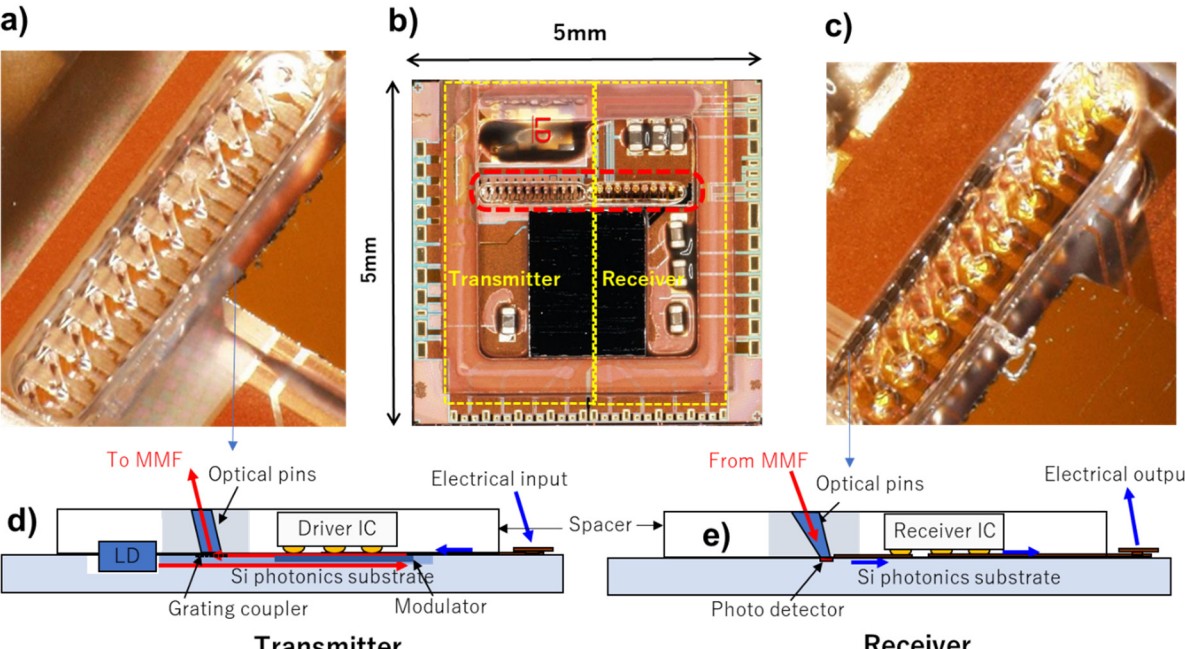

**Figure 4.** IOCore micro-transceiver: (**a**) photo of proprietary transmitter multimode optical interface [3]; (**b**) top photo of complete IOCore micro-transceiver; (**c**) photo of proprietary receiver multimode optical interface; (**d**) cross-section of transmitter section with untapered waveguides ("optical pins"); (**e**) cross-section of receiver section with tapered waveguides ("optical pins").

In the receiver section (Figure 4e), optical signals are coupled from multimode fibers into optical pins on the same array datum as the transmitter optical pins. The receiver optical pins, however, are tapered to both confine and guide the light directly to integrated silicon germanium photodetectors arranged under the optical coupling interface area. This eliminates the requirement for intermediary fiber-to-waveguide coupling structures, such as lossy polarization splitting grating couplers, and moves the light straight to the photodetector, which is directly connected to transimpedance amplifier (TIA) and limiting amplifier (LA) ICs, thus converting the received optical signals into differential voltages.

The optical pin arrays in the transmitter (Figure 4a) and receiver (Figure 4c) sections are embedded in a glass plate mounted onto the silicon substrate. These form the multimode optical interface, which allows bidirectional communication between a multimode fiber array, such as an MT ferrule, and the micro-transceiver, thus permitting a lower cost and more robust assembly solution with respect to temperature and vibration. This differentiates IOCore from most other commercial silicon photonic transceivers, which have single-mode interfaces.

Simulations by AIO Core predict the insertion loss of the transmit optical pin—i.e., the insertion loss between the vertical grating coupler and the output facet of the optical pin at the point of contact with the MMF—to be 0.54 dB. Furthermore, simulations predict the insertion loss on the receiver optical pin—i.e., the insertion loss between the MMF and the receiving photodetector—to be 0.68 dB.

### 2.2. Laser Integration, Reliability and Performance

### 2.2.1. External Light Source vs. Integrated Light Source

The reliability of the laser source is the most important requirement for the high-temperature operation of silicon photonic optical transceivers, yet the integration of reliable light sources is also one of the most challenging aspects of silicon photonic optical transceivers. The three main options for optical sources are (a) an integrated laser; (b) a board-mounted source, but external to the transceiver; and (c) a front-pluggable laser source, as shown in Figure 5. In order to minimize power consumption and cost, the

laser should be integrated into or onto the silicon photonic chip as shown in Figure 5a. The disadvantage of this is that the laser is not field-replaceable and therefore must be extremely reliable in the densely populated component environment of the chip. In order to reduce the risk of transceiver failure due to failure of the laser source, many silicon photonics solutions opt for an external light source (ELS) scheme, whereby the CW laser is not integrated but rather located outside the transceiver assembly. This allows the laser to be in a more temperature-controlled environment. Some higher margin solutions may use Peltier coolers to ensure low-risk and stable operation of the laser source, but uncooled on-board (Figure 5b) or front-pluggable (Figure 5c) modules are likely to become more commercially viable, especially if they are field-replaceable.

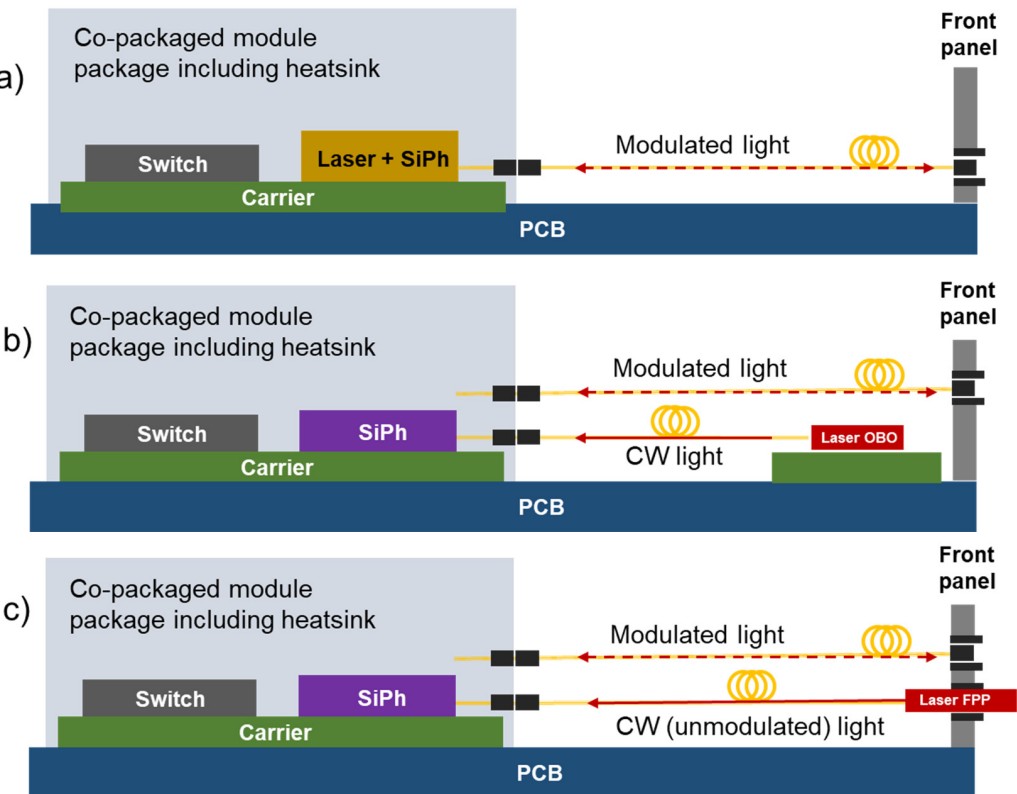

**Figure 5.** Silicon photonics chiplets in a co-packaged assembly with (**a**) integrated laser source, (**b**) external laser source mounted on the board and (**c**) front-pluggable and completely field-replaceable external light source on faceplate.

Figure 6 compares a multimode silicon photonic micro-transceiver with an integrated laser (Figure 6a) to a single-mode silicon photonic transceiver with an external laser source (Figure 6b). External laser source systems typically require polarization-maintaining fibers (PMFs) to ensure low loss coupling of CW light to the transceiver modulator and will include the additional components of a second package for the laser; therefore, they represent a more expensive solution with higher power consumption.

### 2.2.2. Quantum Dot Fabry-Perot Laser Diode Performance at High Temperature

In order to accommodate an integrated laser solution, the laser must be exceptionally reliable at higher temperatures. For this purpose, a Quantum Dot Fabry-Perot Laser Diode (QD-LD) with an operational wavelength of 1310 nm was integrated in the IOCore micro-transceiver.

As shown in the L-I curves of Figure 7a, the QD-LD has a characteristically stable optical output even at high temperatures, with little deviation in output power over a wide temperature range. In addition, as shown in the RIN characteristics of Figure 7b, the QD-LD exhibits strong suppression of noise due to back-reflected light; therefore, they can

be coupled into a silicon photonic circuit without an isolator, allowing further packaging and cost reduction.

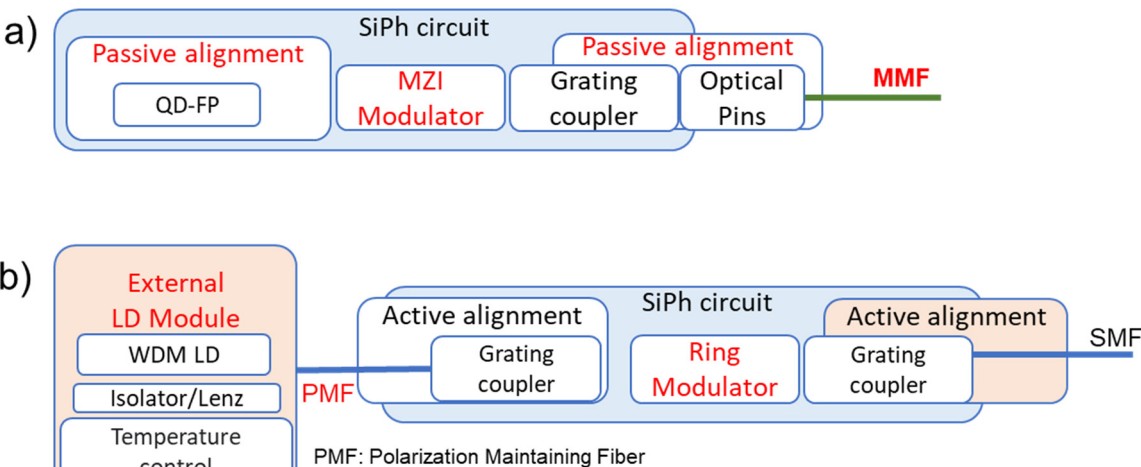

**Figure 6.** Functional block comparison of different laser integration schemes. (**a**) Integrated laser scheme in single wavelength IOCore micro-transceiver with multimode fiber; (**b**) high-end WDM silicon photonics transceiver with external multiwavelength laser source.

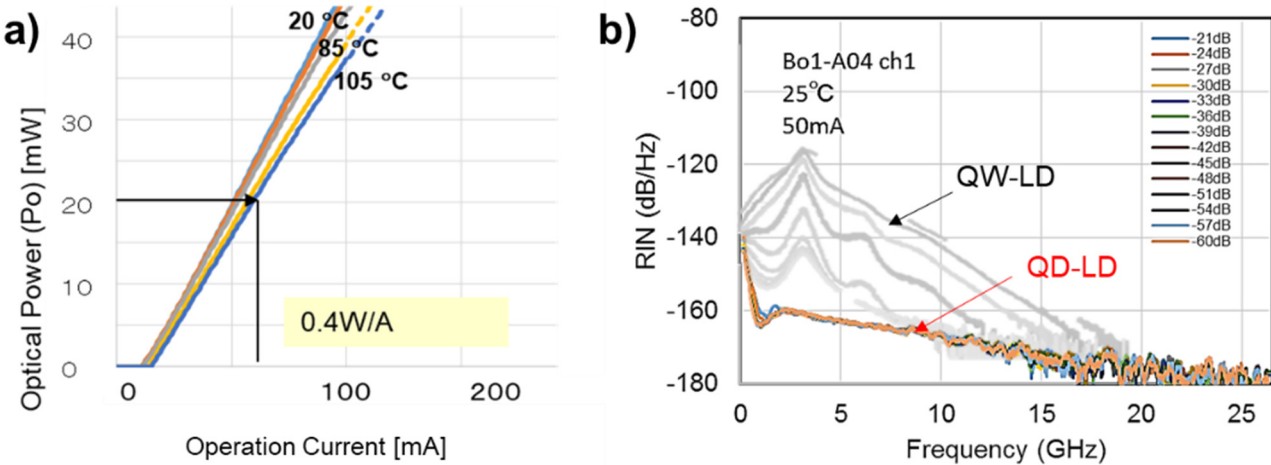

**Figure 7.** QD-LD performance: (**a**) L-I characteristics of QD-LD at different temperatures; (**b**) RIN for back reflection.

### 2.2.3. Redundant Laser Scheme for Increased Reliability

The first-generation IOCore micro-transceiver incorporated a single QD-LD with four CW 1310 nm outputs, each of which was modulated onto each transmitter channel directly (Figure 8a). However, this provided the lowest reliability as any single failure in any one of the outputs or the laser module would have resulted in the failure of the overall system.

In order to address this, a redundant optical circuit was introduced in the second-generation micro-transceiver (Figure 8b), wherein the original single four-channel QD-LD array was replaced by two independent two-channel QD-LD arrays. One QD-LD array is the primary device, with both its optical outputs passed to 3 dB power couplers which split the light equally into two branches, resulting in four CW streams, each of which is modulated onto a transmitter channel. The second QD-LD array is held on stand-by, only to be activated upon the failure of the primary QD-LD array. An additional Mach–Zehnder switch was designed between the outputs of both the primary and secondary QD-LD arrays and the transmitter channels. As shown in Figure 8b, in the event of failure of the primary QD-LD array, the secondary QD-LD array is activated and its two outputs are switched to the 3 dB couplers, which in turn split them into four transmitter channels.

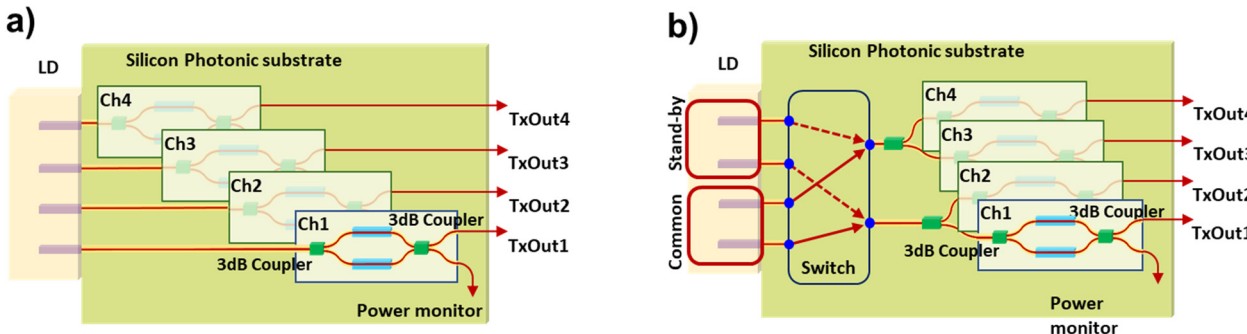

**Figure 8.** IOCore laser redundancy circuit: (**a**) conventional silicon photonic circuit scheme with single QD-LD array with four optical outputs; (**b**) second-generation IOCore micro-transceiver with two QD-LD arrays, each with two CW optical outputs [3].

Using the parameters obtained from reliability testing of the QD-LD, the time within which the chance of failure was 1% was increased from 10,000 to 100,000 h when moving from the non-redundant laser circuit scheme to the redundant scheme.

The qualification of the redundancy circuit scheme is underway and expected to be deployed in commercial devices by the start of 2022.

### 2.3. Passive Assembly of QD-LD into Silicon Photonics Chip

As shown in Figure 9a, the QD-LD array (Figure 9b) is passively mounted onto the silicon photonics substrate with silicon pedestals providing vertical alignment, and a fiducial is formed on the substrate to enable visual placement with respect to corresponding alignment marks on the QD-LD chip (Figure 9c) enabling horizontal alignment. Once the QD-LD is aligned, AuSn solder is applied to fix the array in place.

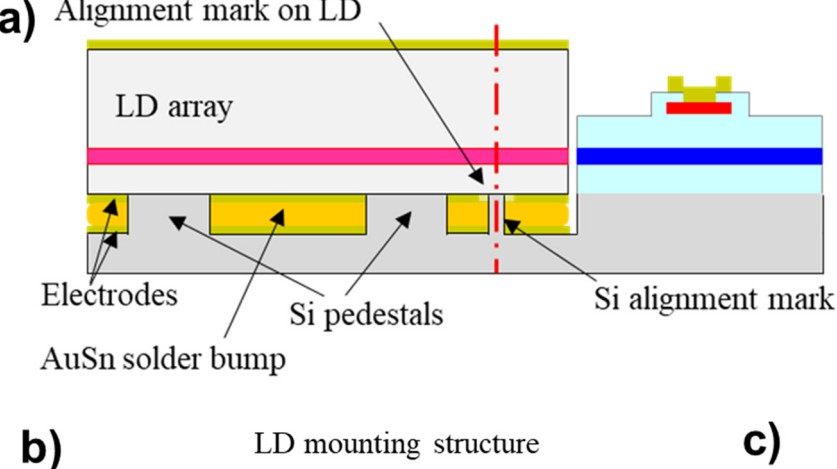

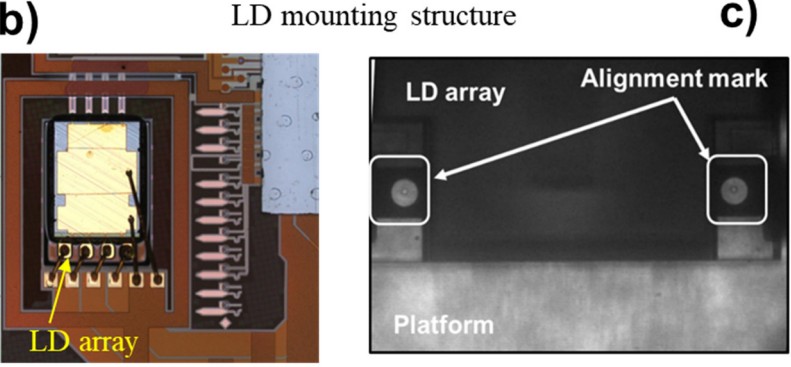

**Figure 9.** Passive assembly of QD-LD onto silicon photonics substrate. (**a**) Side cross-sectional schematic showing passive assembly features of QD-LD array onto silicon photonics substrate; (**b**) photo of QD-LD array; (**c**) photo of alignment marks used to visually align QD-LD array.

This assembly process can be automated with a pick-and-place machine, allowing high-volume, high-yield production.

Figure 10a shows the statistical distribution of mounting accuracy before and after solder attachment, with 99% (σ = 2.3) of mountings falling within an accuracy of ±3 μm.

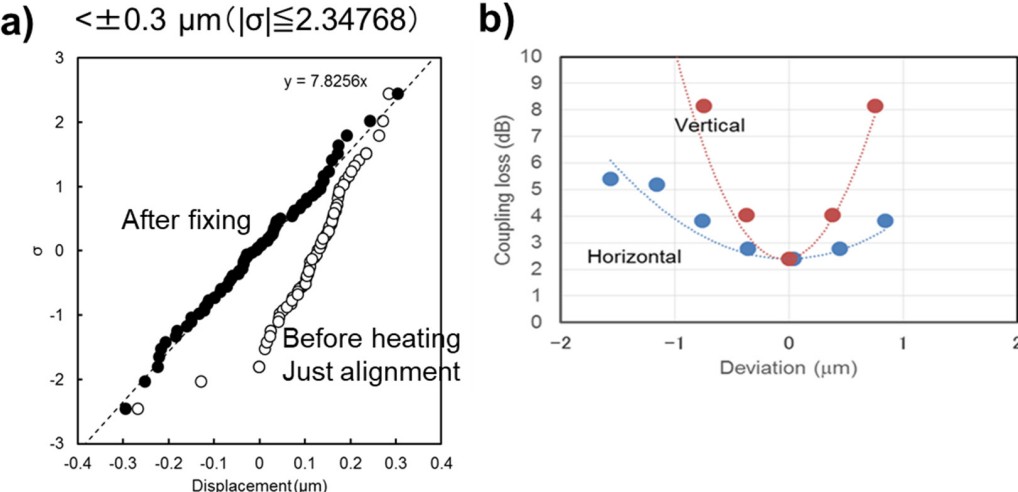

**Figure 10.** Laser mounting accuracy and coupling loss. (**a**) Distribution of mounting accuracy; (**b**) coupling loss variation with horizontal and vertical misalignment.

The QD-LD aperture is aligned to a tapered spot size converter (SSC) in the silicon photonic substrate, which allows laser light to couple to the silicon waveguides. Figure 10b shows the coupling loss variation with horizontal and vertical misalignment. The minimum coupling loss between QD-LD laser aperture and the silicon waveguide was measured to be 2 dB. The vertical misalignment tolerance is tighter than the horizontal misalignment tolerance, with a vertical misalignment of 0.5 μm resulting in ~4 dB coupling loss, while a horizontal misalignment of 0.5 μm resulted in 3 dB coupling loss.

As described in Section 2.8, changes in temperature give rise to a change in the emission wavelength of the QD-LD, which in turn will also affect the coupling loss, depending on the wavelength dependence of the spot size converter.

### 2.4. Mach–Zehnder Modulator Design for High-Temperature Operation

Figure 11a shows the schematic design of the silicon photonic Mach–Zehnder modulator (MZM) with distributed push–pull phase change nodes along each arm. The modulator was designed to provide on-off keying (OOK) over a wide temperature and frequency range. Figure 11b shows the frequency roll-off characteristics of the modulator. IOCore operation at 25 and 32 Gbps NRZ per channel has been reported [3], and the development of 50 Gbps NRZ per channel capability is underway.

### 2.5. Optical Output Power

Figure 12 shows the optical output characteristics (OMA) of the IOCore micro-transceiver as measured from the Tx optical pin. When the drive current is fixed at 75 mA, an average optical output of 0.14 dBm is measured, but by adjusting the drive current, the optical output can be increased to +1 dBm when in use.

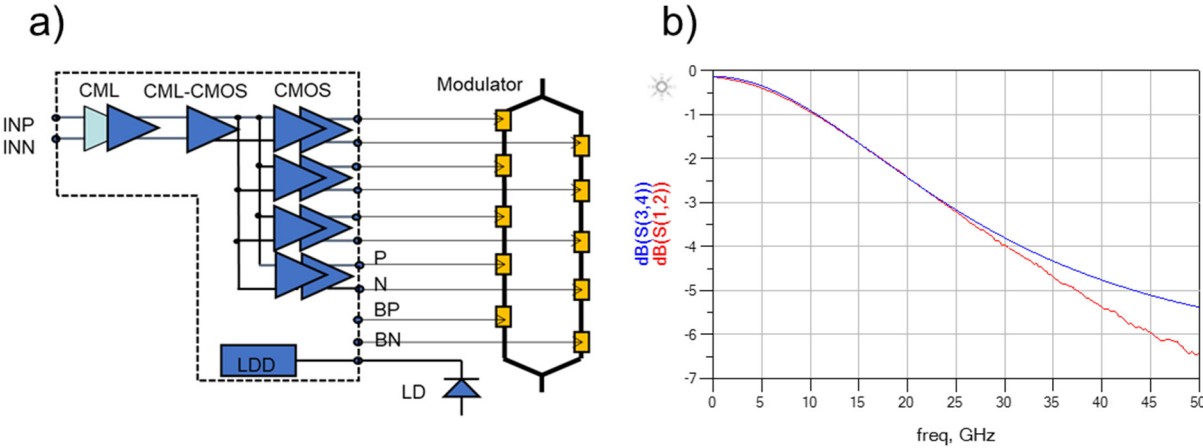

**Figure 11.** Mach–Zehnder modulator (MZM) design and performance. (**a**) Schematic diagram of MZM driven by CMOS IC; (**b**) frequency characteristics of the MZM.

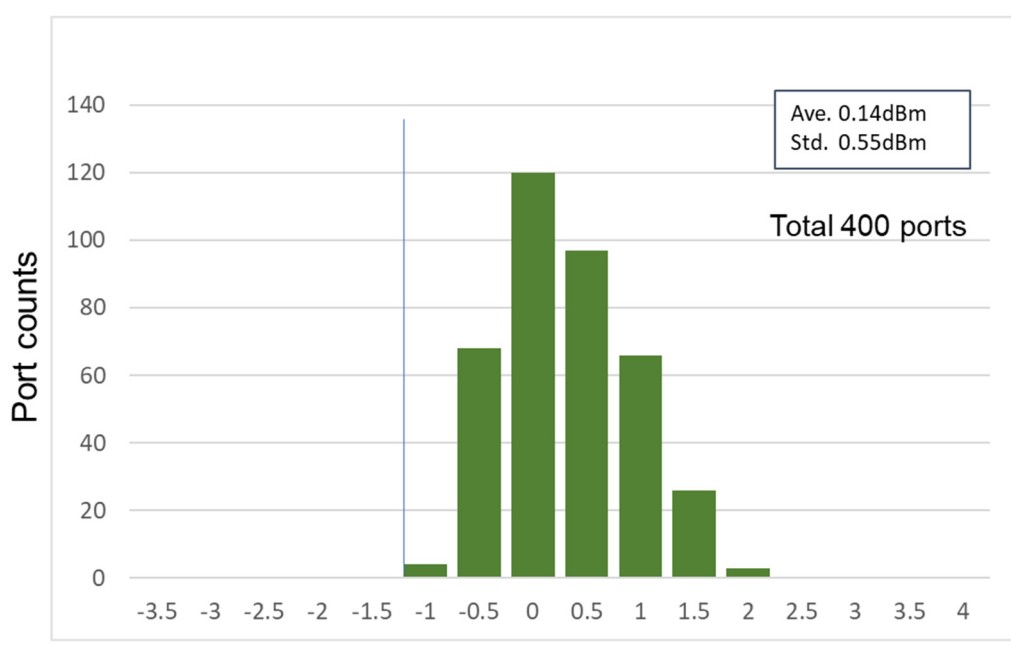

**Figure 12.** Distribution of optical modulation amplitude as measured on the outputs of the IOCore Tx optical pins.

### 2.6. Optical Transmitter Signal Integrity Measurements

Figure 13 shows the optical eye diagrams from the four output channels driven at 25.78 Gbps. The eye diagrams in the left column were recorded at an ambient temperature of 25 °C, while the eye diagrams in the right-hand column were recorded at an ambient temperature of 105 °C. The IOCore micro-transceiver does not include a clock and data recovery (CDR) circuit; nonetheless, a negligible difference in signal integrity and low jitter were recorded across the temperature range.

Transceivers with data rates of 25 and 32 Gbps have been reported [3,19], and a 50 Gbps version is currently under development with demonstration planned in 2022.

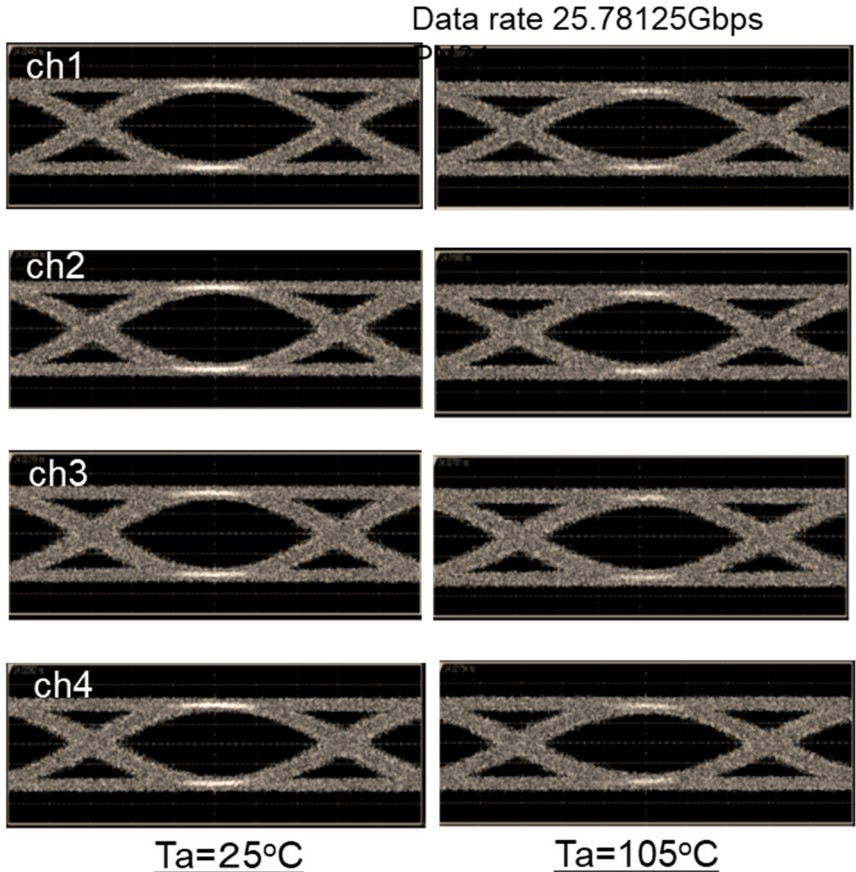

**Figure 13.** Optical eye diagrams on all four channels at 25 and 105 °C.

*2.7. Optical Receiver Performance at 105 °C*

Figure 14 shows the optical receiver sensitivity profile. The responsivity of the integrated germanium photodetector is 0.5 A/W with a frequency response of 18 GHz. The receiver IC has a gain adjustment function for high-temperature operation, and the degradation in receiver sensitivity is negligible at high temperatures.

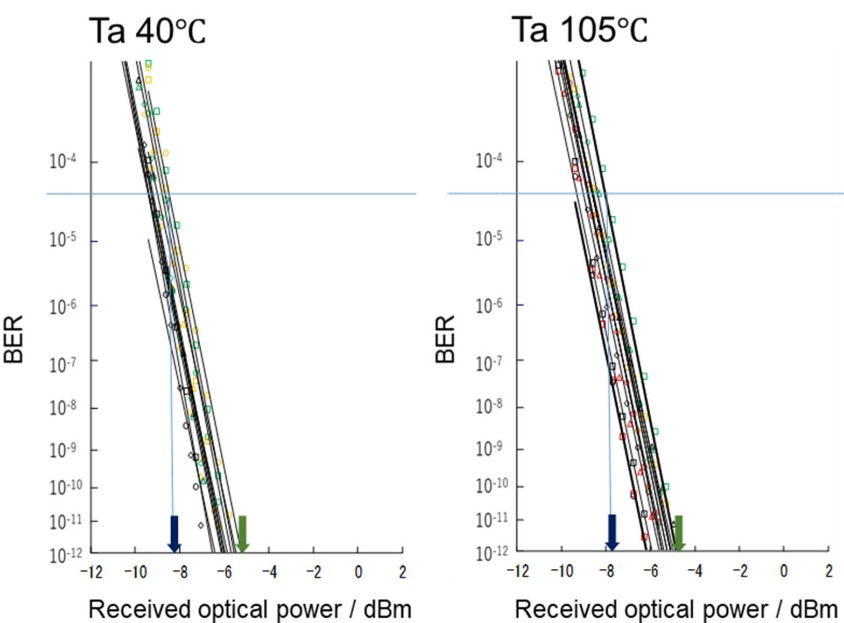

**Figure 14.** Optical bit error rate vs. received optical power.

Table 1 shows that with IEEE 802.3 bm-compliant forward error correction (FEC), reception sensitivity for a BER of $\geq 5 \times 10^{-5}$ can be achieved with received optical power $> -8.2$ dBm at an ambient temperature of 40 °C, increasing slightly to $-7.8$ dBm at an ambient temperature of 105 °C, while without FEC applied, a BER of $\geq 10^{-12}$ can be achieved with a received optical power of $-5.6$ dBm at an ambient temperature of 40 °C, increasing slightly to $-5.0$ dBm at an ambient temperature of 105 °C.

**Table 1.** Reception sensitivity.

|  | **Ta 40 °C (Tj 60 °C)** | **Ta 105 °C (Tj 125 °C)** |
|---|---|---|
| w/ FEC | $-8.2$ dBm | $-7.8$ dBm |
| w/o FEC | $-5.6$ dBm | $-5.0$ dBm |

*2.8. Multimode Optical Interface Performance Challenges at High Temperature*

In the transmitter section, the modulated light is coupled through a vertical grating coupler into a tilted waveguide called an optical pin (Figure 15a). The emission angle of the grating coupler varies with wavelength (Figure 15b), and quantum dot lasers exhibit a wavelength shift with temperature (Figure 15c); this in turn results in a change in the direction of optical output over the temperature range. In the case of Fabry-Perot lasers, the wavelength has a temperature dependence of ~0.56 nm/°C, so when the temperature rises from 25 to 105 °C, the emission angle of the grating coupler changes by about 5° due to the corresponding variation in wavelength.

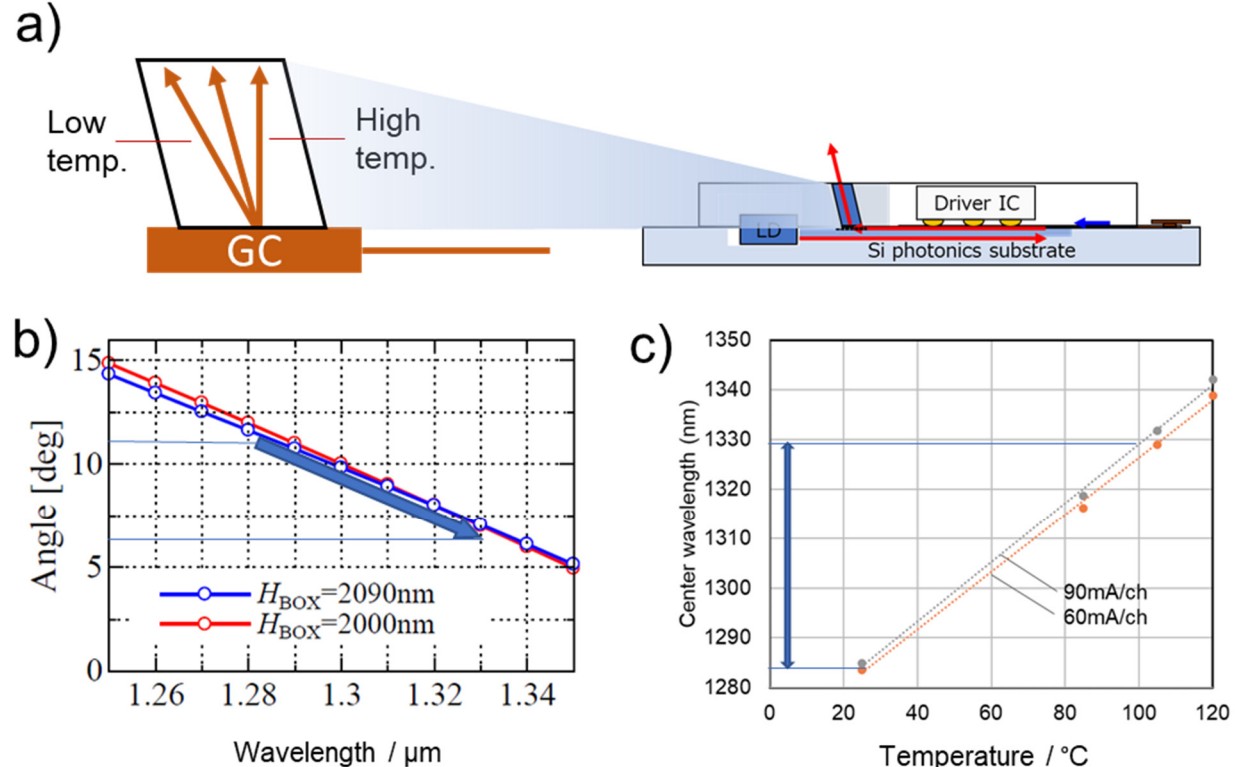

**Figure 15.** Optical coupling interface performance over temperature. (**a**) Schematic view of optical pin on Tx side accommodating variation in directional angle from the grating coupler over temperature; (**b**) variation of angle with wavelength; (**c**) variation of wavelength with temperature.

If the multimode optical fiber were directly connected to the grating coupler, the coupling loss would change more strongly over the temperature range in response to the change in emission angle. However, the optical pin has a higher NA of 0.4 and a core diameter of 35 μm, which confines the light more strongly in the optical pin core than a multimode fiber would and can thus better accommodate this variation in emission angle.

This intermediary waveguide then couples light into a multimode fiber with little variation in coupling loss over the temperature range.

### 2.9. Electro-Optical Module for High-Temperature Operation

The IOCore micro-transceiver is a building block that can be integrated into different types of electro-optical modules or co-packaged configurations. Figure 16b shows a low-form factor electro-optically pluggable module called the EOB (Electro-Optical Blade), which includes advanced heat dissipation structures on the upper and lower surfaces of the IOCore micro-transceiver, enabling operation at 105 °C. When used at an ambient temperature of 105 °C, the junction temperature, Tj, is designed to be 120 °C or less, which is within the temperature specifications for the ICs and laser.

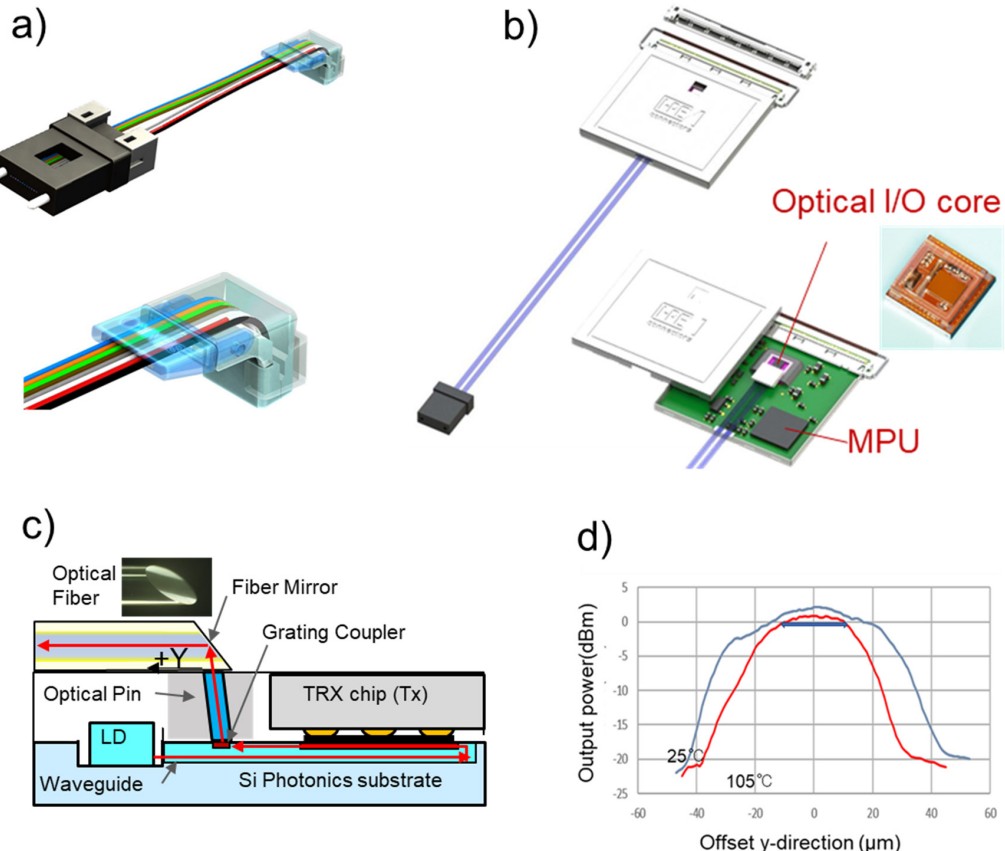

**Figure 16.** (**a**) Commercial vertical fiber array unit (courtesy of Senko); (**b**) electro-optically pluggable micro-transceiver module "EOB"; (**c**) cross-sectional schematic showing low-form factor coupling based on cleaved fiber; (**d**) misalignment tolerance of cleaved fiber at 25 to 105 °C.

### 2.10. Low-Form Factor Optical Coupling Interface

Commercial vertical multimode fiber array units, as shown in Figure 16a, can be used to couple to the optical interface; however, such units take up an additional height of at least 3.8 mm today and would therefore be unsuitable for many space-constrained applications such as co-packaged optics in a 5G+ environment or data centers.

The EOB instead uses a fiber with a cleaved angled end facet (Figure 16c) allowing light to be coupled directly from the optical pin to a horizontally arranged fiber, thus enabling very low profile interfaces. The 1 dB misalignment tolerance of the cleaved fiber along the axis of the horizontal fiber (denoted as offset y-direction) is about ±10 μm (Figure 16d) at 25 °C and is slightly tighter at around ±8 μm at 105 °C.

## 3. Discussion

In this paper we have primarily addressed the performance of the micro-transceiver in high-temperature environments, including, in particular, the channel speed, total throughput and low power consumption, where increasing the data rate per channel reduces the power consumption per bit of optical interconnect. It should be noted that the micro-transceiver is particularly very well suited to integration in board-mountable and co-packaged optical modules, which are currently of great importance to hyperscale data centers, and this has been reported by Kurata et al. [3].

In addition to these important performance metrics, we also focused on the required operating conditions. Here, in particular, there is a need for high-temperature operation combined with high reliability which exceeds that of pluggable components by an order of magnitude and enables a substantially lower cost solution than conventional single-mode silicon photonic devices. In order to meet these requirements, we have proposed a silicon photonic micro-transceiver with a multimode interface. Stable operation, without the need for temperature control systems, is possible at temperatures up to 105 °C. Furthermore, the introduction of a redundant laser circuit ensures over an order of magnitude higher reliability than a system with a non-redundant laser circuit. Thus, the reported micro-transceiver meets two of the key requirements outlined above.

To achieve the all-important low cost point, we need to consider both the device and mounting/assembly costs. On the device side, a reduction in the chip size is crucial as it enables a lower cost than conventional devices. This is of particular importance for high-volume applications such as optical interconnects, where small cost differences will quickly scale. On the packaging side, we can achieve low-cost assembly through the use of a multimode interface, which eliminates the high costs associated with high-precision single-mode fiber-to-chip assembly of conventional silicon photonics transceivers. The final important factor affecting the device cost is the fabrication and assembly yield. We have presented a micro-transceiver with very robust processing and assembly schemes, enabling the requisite high yield.

Although it is possible to implement Tbps with more channels in a single micro-transceiver chip by applying this technology, we are actively working towards transceivers with aggregate data rates above 1 Tbps whereby multiple "standard" four-channel micro-transceivers are mounted into different module form factors. Since the four-channel micro-transceiver chip size is small, many chips can be incorporated into standard module packages, allowing bandwidths to be scaled. This allows modules with n x four-channel micro-transceiver building blocks to address a wide diversity of applications beyond 5G and 6G, where n can be anything from '1' (e.g., for internal optical communication in automotive applications) to '64' (and beyond) for co-packaged optical ASICs in hyperscale data centers.

The development of 128 Gbps ($4 \times 32$ Gbps/ch) transceivers with the same configuration as the $4 \times 25$ Gbps/ch transceivers reported in this paper is planned to be completed before 2022, and the development of 50 Gbps NRZ/ch 200 Gbps chips and 2.4 Tbps modules is in progress.

In the future, optimization of the optical wiring system in housing will become an important consideration in the realization of micro-transceivers. Here, we anticipate the use of multimode optical wiring because of its ease of connection, low cost and increased reliability, but the transmission medium and optical connector present areas of further research and development.

## 4. Conclusions

In this paper, we have reported on the design and performance of the IOCore multimode micro-transceiver and its viability for deployment in large-scale integration or co-packaged assemblies in 5G+ and other high-temperature, space-constrained environments. One important feature of the second-generation IOCore micro-transceiver is the integrated quantum dot laser and laser redundancy circuit, which enable operation in high-

temperature environments with substantially improved reliability. The relaxed assembly constraints of a multimode interface give rise to a reduced-cost solution at the expense of restricting the device-to-device optical link distance to under 500 m. This trade-off allows multimode silicon photonics micro-transceivers to satisfy very short-reach, low-cost, temperature-resilient requirements in diverse applications. In addition, the IOCore micro-transceiver has been evaluated under single- and dual-phase immersion cooling environments, with a negligible change in performance compared to operation in a standard air-cooled environment as reported in [19].

**Author Contributions:** Conceptualization, K.K., K.N., F.C. and L.G.; methodology, K.K., K.N., L.G. and F.C.; validation, K.N., Y.H., K.Y., T.M., S.K. and M.K.; formal analysis, K.K., M.K. and S.K.; investigation, K.N., Y.H., K.Y., T.M., S.K., M.K., L.G. and F.C.; writing—original draft preparation, R.P., K.K., L.G. and F.C.; writing—review and editing, L.O., S.A.S. and R.P.; project administration, K.K.; supervision, K.K.; funding acquisition, K.K. All authors have read and agreed to the published version of the manuscript.

**Funding:** This paper incorporates results obtained from the project JPNP20017, commissioned by the New Energy and Industrial Technology Development Organization (NEDO) in Japan.

**Institutional Review Board Statement:** Not applicable.

**Informed Consent Statement:** Not applicable.

**Data Availability Statement:** The data presented in this study are available on request from the corresponding author. The data are not publicly available due to confidentiality restrictions.

**Conflicts of Interest:** The authors declare no conflict of interest.

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
