# Peer review of "Silicon Photonic Micro-Transceivers for Beyond 5G Environments"

_applsci, doi:10.3390/app112210955_

Round 1

Reviewer 1 Report

All references to hyperlinks must indicate access date;

Fig. 12 must be clarified; optical power (dBm) or Optical Modulation Amplitude (OMA) in dB ?

line 329:  "The sensitivity ..." should be replaced by "The Responsivity ..."

Fig. 14 same comment as in Fig. 12; the x scale is in dBm so it is ROP and not OMA

Author Response

The authors thank the reviewer for taking the time to scrutinise this paper and provide valuable feedback, which has been addressed in full as detailed below.

All references with hyperlinks were reconfirmed on 5th November 2021 and the access dates added or updated to accordingly.

Figure 12 and its caption have been amended to clearly state that Optical Modulation Amplitude (OMA) rather than optical power is shown in the figures.

On line 329 “sensitivity” was replaced by “responsivity”

In figure 14 “OMA” has been replaced by “Received optical power / dB”

Reviewer 2 Report

The article reports on the performances of a packaged silicon photonic micro-transceiver for short reach interconnect. The main feature is the use of an integrated quantum dot laser which enable operation in high temperature environments. Another important feature of the presented transceiver concerns the assembly: the passive assembly of the laser on the silicon substrate and the multimode optical interface at the output of the transceiver.

Overall, the paper is clear, interesting and well written. Some references are missing for a better understanding of the various components of the transceiver, especially concerning the glass optical pin which is not a common device in silicon transceivers.

Going into details here are the few minor points to address before publication:

Line 110: typo in “variation in temperate are particularly…”

Line 163: a reference or a detailed explanation describing the optical pin is missing

Line 213: typo in “comsumptive”

Paragraph 2.3: can the author cite a reference for this assembly technique?

Figure 12: I believe the unity of the X axis should be dBm instead of dB.

Figure 14: the unity of the X axis is missing

Paragraph 2.8: The authors should indicate the value of the coupling loss between the chip and the multimode fiber

Author Response

The authors thank the reviewer for taking the time to scrutinise this manuscript and provide valuable feedback, which has been addressed as detailed below. The glass optical pin array is a proprietary component so some detail has been excluded, however additional references have been added to earlier papers by AIO Core, which provide a little more detail.

On line 110 “temperate” has been changed to “temperature”

The text throughout the document has been amended to clarify that the optical pin interface is proprietary and more references have been added to an earlier paper. Though the paper was already in the reference list, more pointers to this reference have been added in the areas where the multimode interface is alluded to.

On line 213, the sentence has been changed to “… therefore they represent a more expensive solution with higher power consumption” and the word “consumptive” removed.

The technique of passive assembly of the QD laser into the silicon photonics platform is disruptive as it allows low cost, high volume laser integration, however it also proprietary to AIO Core so further detail cannot be included at this time.

The unit of the x-axis in Figure 12 has been corrected to dBm.

The x-axis description in Figure 14 has been changed to “Received optical power / dBm”

The value of coupling loss between the chip and multimode fibre in both directions has been added to section 2.1.